# Predictive coding in musical anhedonia: A study of groove

**Peter Benson**[1,2], **Nicholas Kathios**[3], **Psyche Loui**[1,3]*

**1** Dept. of Music, College of Arts, Media, and Design, Northeastern University, Boston, Massachusetts, United States of America, **2** Dept. of Computer Science, Khoury College of Computer Sciences, Northeastern University, Boston, Massachusetts, United States of America, **3** Dept. of Psychology, College of Science, Northeastern University, Boston, Massachusetts, United States of America

* p.loui@northeastern.edu

**Data Availability Statement:** All stimuli, data, and code used for this study are publicly available from the OSF repository (https://osf.io/62a9v/).

**Funding:** Supported by NSF-CAREER 1945436, NSF-BCS 2240330, NIH R01AG078376 and R21AG075232 to PL, NSF GRFP award (DGE-

## Abstract

Groove, or the pleasurable urge to move to music, offers unique insight into the relationship between emotion and action. The predictive coding of music model posits that groove is linked to predictions of music formed over time, with stimuli of moderate complexity rated as most pleasurable and likely to engender movement. At the same time, listeners vary in the pleasure they derive from music listening: individuals with musical anhedonia report reduced pleasure during music listening despite no impairments in music perception and no general anhedonia. Little is known about musical anhedonics' subjective experience of groove. Here we examined the relationship between groove and music reward sensitivity. Participants (n = 287) heard drum-breaks that varied in perceived complexity, and rated each for pleasure and wanting to move. Musical anhedonics (n = 13) had significantly lower ratings compared to controls (n = 13) matched on music perception abilities and general anhedonia. However, both groups demonstrated the classic inverted-U relationship between ratings of pleasure & move and stimulus complexity, with ratings peaking for intermediately complex stimuli. Across our entire sample, pleasure ratings were most strongly related with music reward sensitivity for highly complex stimuli (i.e., there was an interaction between music reward sensitivity and stimulus complexity). Finally, the sensorimotor subscale of music reward was uniquely associated with move, but not pleasure, ratings above and beyond the five other dimensions of musical reward. Results highlight the multidimensional nature of reward sensitivity and suggest that pleasure and wanting to move are driven by overlapping but separable mechanisms.

## Introduction

Why does music move us? The ubiquitous observation that humans are inclined to move to music has inspired the evolutionary argument that musicality evolved for social bonding: by having rhythm and inspiring people to move in synchrony, music brings people together [1]. The psychological construct of groove, defined as the pleasurable urge to move to music, has become established as an area of study that offers a window into the relationship between

1938052) to NK, and internal seed funding from
Northeastern University.

**Competing interests:** The authors have declared
that no competing interests exist.

emotion and cognition. What kinds of rhythmic structure engender pleasurable movement,
and for whom? Developing an empirical understanding of groove requires spanning multiple
levels of analysis, from phenomenological to neural [2]. This empirical understanding may in
turn offer rehabilitative possibilities through music-based interventions for movement disorders such as Parkinson's Disease [3, 4].

In the first psychophysical study on groove, participants listened to rhythmic drum patterns
and rated the extent to which each one gave them the experience of pleasure and made them
want to move. These rhythmic drum patterns varied in complexity, ranging from highly regular (low-complexity) to highly irregular (high-complexity). Results showed that syncopated
drum patterns, operationalized as medium in rhythmic complexity, are rated as most pleasurable and most likely to engender movement [5]. This inverted-U-shaped relationship between
rhythmic complexity and movement, and between rhythmic complexity and pleasure [6–9]
have informed the predictive coding model for music (PCM, [10]): According to the PCM
model, groove is linked to precision-weighted prediction errors that music engenders over
time, and the brain learns to build an internal model of the world by minimizing these prediction errors [8, 10]. In a Western context, for example, given that most listeners have experience
through implicit statistical learning processes to binary metric structure–with alternating
strong and weak beats recurring over time–predictions would be most certain for notes falling
on the first (strongest) beat of a phrase, followed by the second (weaker) beat, followed by
notes that do not fall on a beat at all. Syncopated rhythms are patterns that avoid the strongest
predictions, while not being too complex to learn.

In considering the relationship between music, learning, and emotion, it is also clear that
significant individual variability exists in the general population. Responses on the extended
Barcelona Music Reward Questionnaire (eBMRQ; [11, 12]) show that musical reward is multidimensional, typically falling into six factors: sensorimotor reward (moving to music), social
reward (sharing music with others), music-seeking (finding new music), emotion evocation,
mood regulation, and absorption into music. While most people find music to be rewarding in
these ways, people with musical anhedonia report an insensitivity to the rewards of music listening across these dimensions [13, 14]. Specific musical anhedonia is defined as a selective
lack of pleasurable responses to music, despite normal hedonic responses to other sensory and
aesthetic stimuli, as well as normal auditory perceptual abilities [15–17]. Individuals with
musical anhedonia have disrupted structural and functional connections between auditory
and reward systems, as shown in structural and functional neuroimaging studies coupled with
behavioral testing [14, 15, 18, 19].

Recent research in musical anhedonia has highlighted a difficulty in this population in mapping musical predictions to reward [20]. Using tonal predictions generated *de novo* in the Bohlen-Pierce scale, it was found that individuals with musical anhedonia showed similar ratings
as matched controls in predictions, operationalized as familiarity ratings; however their liking
ratings were lower than controls and did not follow the same patterns as predicted from familiarity ratings. This suggests that musical anhedonics may have difficulty with predictive coding, specifically in mapping tonal predictions to reward. While this result links the study of
musical anhedonia to predictive coding, the predictions being manipulated were tonal only.
Little is known about how musical anhedonia might affect the pleasurable urge to move, or the
preference for complexity in the domain of rhythm. It is also not clear how the multiple
dimensions of individual differences in musical reward might relate to these experiences of
pleasure and wanting to move to music.

Here we test the overall hypothesis that the relationship between perceived complexity and
the pleasurable urge to move is disrupted in people with musical anhedonia. More specifically,
our hypotheses are twofold: Firstly, we expect that individuals with specific musical anhedonia

would show less of an inverted-U relationship between complexity and pleasure, as well as lower pleasure ratings overall, compared to controls matched for music perception abilities and general anhedonia. Secondly, we expect that individual differences in musical reward sensitivity would be a significant predictor of preference for complexity, as defined by a significant interaction between individual differences in music reward sensitivity and perceived complexity on the outcome variables of pleasure and move ratings.

## Methods

### Participants

A total of 288 English-speaking participants from the United States were recruited through Prolific between January and May, 2023. These participants were selected from another online study [20]. One participant was removed from analyses due to providing incomplete data (resulting in a total n of 287; see Table 1 for demographic information). To determine the sample size, our initial constraint was that we were hoping to recruit comparably-sized samples of musical anhedonics and controls matched for PAS scores and MET scores. Given that previous publications on musical anhedonia had used samples of n = 10–13 per group [15, 16], and given that we expect ~5% of the overall sample to test within the range of musical anhedonia [11], we opted to recruit 20 times the target n of 13 per group to reach a sample of at least n = 260. We checked for the power of the current sample size to detect a main effect of rhythmic complexity using openly available data from [21]. We conducted power analyses with simulations using the R package *simr* [22] to detect a main effect of rhythmic complexity on pleasure ratings using a linear mixed effects model with a random by-participants intercept. The effect size and random effects were determined running an identical model on the data

**Table 1. Mean and standard deviation scores, as well as demographic information, for the identified musical anhedonic and control groups, along scores across the entire study sample.**

| Scale and Survey Measures Across Musical Anhedonics, Matched Controls, and Study Sample | | | |
|---|---|---|---|
| **Scale** | **Musical Anhedonics n = 13 M (SD)** | **Matched Controls n = 13 M (SD)** | **All Participants n = 287 M (SD)** |
| MET Melodic Score | 35.0 (5.11) | 36.85 (4.97) | 35.18 (5.37) |
| MET Rhythmic Score | 37.38 (5.34) | 39.08 (5.0) | 36.48 (5.65) |
| PAS without auditory subscale | 10.54 (3.28) | 10.38 (3.23) | 9.86 (6.38) |
| eBMRQ* | 64.15 (11.17) | 87.46 (8.22) | 91.79 (14.93) |
| Age | 40.31 (19.18) | 33.77 (8.48) | 34.16 (9.04) |
| Sex | 9M, 4F | 5M, 8F | 145M, 140F |
| Gender (%) | | | |
| Man | 69.23% | 38.46% | 50.87% |
| Woman | 30.77% | 61.54% | 47.04% |
| Non-binary/Other | 0% | 0% | 1.39% |
| Choose not to disclose | 0% | 0% | 0.35% |
| Race/Ethnicity (%) | | | |
| White | 84.62% | 69.23% | 68.29% |
| Black or African American | 0% | 7.69% | 9.41% |
| Native American | 0% | 0% | 0% |
| Asian | 15.38% | 0% | 7.32% |
| Native Hawaiian or Pacific Islander | 0% | 0% | 0% |
| Hispanic or Latino | 0% | 7.69% | 5.57% |
| Other/Mixed Race/Ethnicity | 0% | 7.69% | 8.71% |
| No response | 0% | 7.69% | 0.7% |

which had a partial eta squared of 0.21 [21]. Given our design of 5 ratings per condition, the simulated power analysis revealed that 5 participants in each group were required to detect a main effect of rhythmic complexity.

## Stimuli

**Measures of rhythmic complexity.**   To manipulate the rhythmic complexity of musical excerpts, we drew our stimuli from a newly-released corpus of 40 drum pattern excerpts from Western popular music that are normed for perceived complexity [23]. While rhythmic complexity has been operationalized using syncopation [5], listeners' judgments of complexity (i.e., perceived complexity, Senn et al., 2023), and computationally-derived measurements of complexity (i.e., estimated complexity, [24]), we opted to use perceived complexity in the present study. We chose not to use the computationally-derived syncopation index as a proxy for complexity because past work suggests that features beyond acoustic properties (such as participant familiarity and musical experience) can influence the subjective experience of groove [25]. For this reason, we also opted for perceived complexity over estimated complexity because it quantifies complexity as a perceptual feature rather than a computationally-derived one.

**Stimuli selection procedure.**   To allow for similar experimental design and statistical analyses as previous studies on between-group comparisons on groove [26, 27], we chose fifteen of the total forty stimuli from the Senn and colleagues (2023) database, categorized into groups of low, intermediate, or high levels of perceived complexity. To determine these groups, we first split all 40 excerpts into tertiles based on the normed perceived complexity values. Then, the five stimuli with the lowest perceived complexity values (which served as the "Low" complexity group) were matched with five of the closest excerpts on loudness, number of onsets, initial tempo, and duration from the middle ("Intermediate" complexity group, five total stimuli) and upper tertile ("High" complexity group). See S1 Fig and S1 Table in S1 File for values for each stimulus on loudness, number of onsets, initial tempo, duration, as well as all three measures of rhythmic complexity and S2 and S3 Tables in S1 File for averages of these values across the three complexity groups.

## Procedures

Participants completed this experiment on Qualtrics and the online behavioral experiment platform Gorilla. Written informed consent was obtained in accordance with the IRB-approved protocol 18-12-13 at Northeastern University. After consenting, participants were screened using an online headphone check [28]. Following this, participants completed the Musical Ear Test (MET; [29]) on Gorilla [30]. The MET is a sensitive measure of musical abilities that can distinguish between professional, amateur, and non-musicians. The MET consists of a melodic and rhythmic test. In each test, participants are exposed to 52 pairs of musical excerpts (for a total of 104 trials) and are asked to determine whether each pair of excerpts is identical. For trials that contain different excerpts in the melodic subtest, the second excerpt has a single pitch violation. For trials that contain different excerpts in the rhythmic subtest, the second except has a single rhythm change.

Following this, participants were redirected to Qualtrics to complete the main task being reported in this manuscript Participants were exposed to the fifteen drum excerpts selected from the Senn et al. (2023) corpus [23]. Each stimulus was presented in a randomized order across participants. Following past work on groove and pleasure [5], participants were asked to rate each excerpt both on how much it made them want to move (on a scale from 1 meaning "Not at all" to 5 meaning "Very much") and how much pleasure they experienced listening to the excerpt (on a scale from 1 meaning "None" to 5 meaning "A lot").

Participants then completed a battery of psychometric surveys which included the Absorption in Music Scale (AIMS [31]), the Social Reward Questionnaire (SRQ; [32]), the Escapism Scale [33], the Aesthetic Experience Scale in Music (AES-M; AES; [34]), the Healthy-Unhealthy Music Usage Scale (HUMS; [35]), the Trait portion of the State-Trait Anxiety Inventory (STAI; Spielberger, 1970), the Questionnaire of Unpredictability in Childhood (QUIC; [36]), the Confusion, Hubbub, and Order Scale (CHAOS; [37]), the Connor-Davidson Resilience Scale (CD-RISC-10; [38]), and measures of childhood and adolescent exposure to deprivation and threat [39]. Results from these measures will be reported in a separate manuscript [40].

We also use two survey measures collected from an earlier wave of data collection (see "Participants"): the Barcelona Music Reward Questionnaire (BMRQ [11]) and the Revised Physical Anhedonia Scale (PAS [41]). These two scales are often administered together to identify music-specific anhedonics [11, 15, 16]. The BMRQ is a 20-item measure of individual differences in sensitivity to musical reward based on five factors: musical seeking, emotion evocation, mood regulation, sensory-motor, and social reward. Recently, absorption in music has been identified as an additional sixth factor of music reward (on the extended BMRQ, or eBMRQ [12]). As this factor consists of four questions that originate from the AIMS, we calculated participants' total eBMRQ by adding these questions from the AIMS to participants' BMRQ scores.

The PAS is a self-report questionnaire measuring general anhedonia, or inability to experience pleasure. It consists of 61 items which ask participants to identify if statements describing typically pleasurable experiences (i.e. "Trying new foods is something I have always enjoyed") are true. Ten of these items ask specifically about hedonic responses to sounds (both musical and non-musical; i.e. "The sound of organ music has often thrilled me"). Because the purpose of administering this measure was to determine the specificity of musical anhedonia according to eBMRQ scores, auditory items of the PAS were excluded from participants' overall PAS score (as in [15]).

## Analysis plan

**Identifying individuals with musical anhedonia.** To test for differences in experiences of groove between musical anhedonics and matched controls, we first identified individuals with musical anhedonia. These individuals were classified as those who scored below the tenth percentile (<72.6) on the overall musical reward score of the eBMRQ in our sample while also scoring more than one standard deviation above the mean on the PAS (>16.24) after removing items from the auditory subscale [16]. Identified musical anhedonics (n = 13) were then matched with their closest counterpart (who scored above the eBMRQ cut off; n = 13) on music perception abilities (as assessed by both the rhythmic and melodic MET scores) and general anhedonia (as assessed by PAS scores without the auditory subscale). Scores for each group can be found on S4 Table in S1 File (see Fig 1 for the distribution of eBMRQ and PAS scores across the entire sample).

**Between-group comparisons.** With these participants, we then constructed linear mixed-effects models with complexity tertile (i.e. low, intermediate, and high) as a predictor variable of ratings of pleasure and wanting to move (in separate models) using the *lme4* package [42] in R. To test our first hypothesis, we included group membership (musical anhedonics vs. matched controls) as an interaction term in these models. Following past work demonstrating an inverted-U relationship between ratings of pleasure and wanting to move and rhythmic complexity [21], we tested a quadratic contrast on this model. To test for differences in this inverted-U relationship across groups, we extracted interaction contrasts using the *emmeans* package [43].

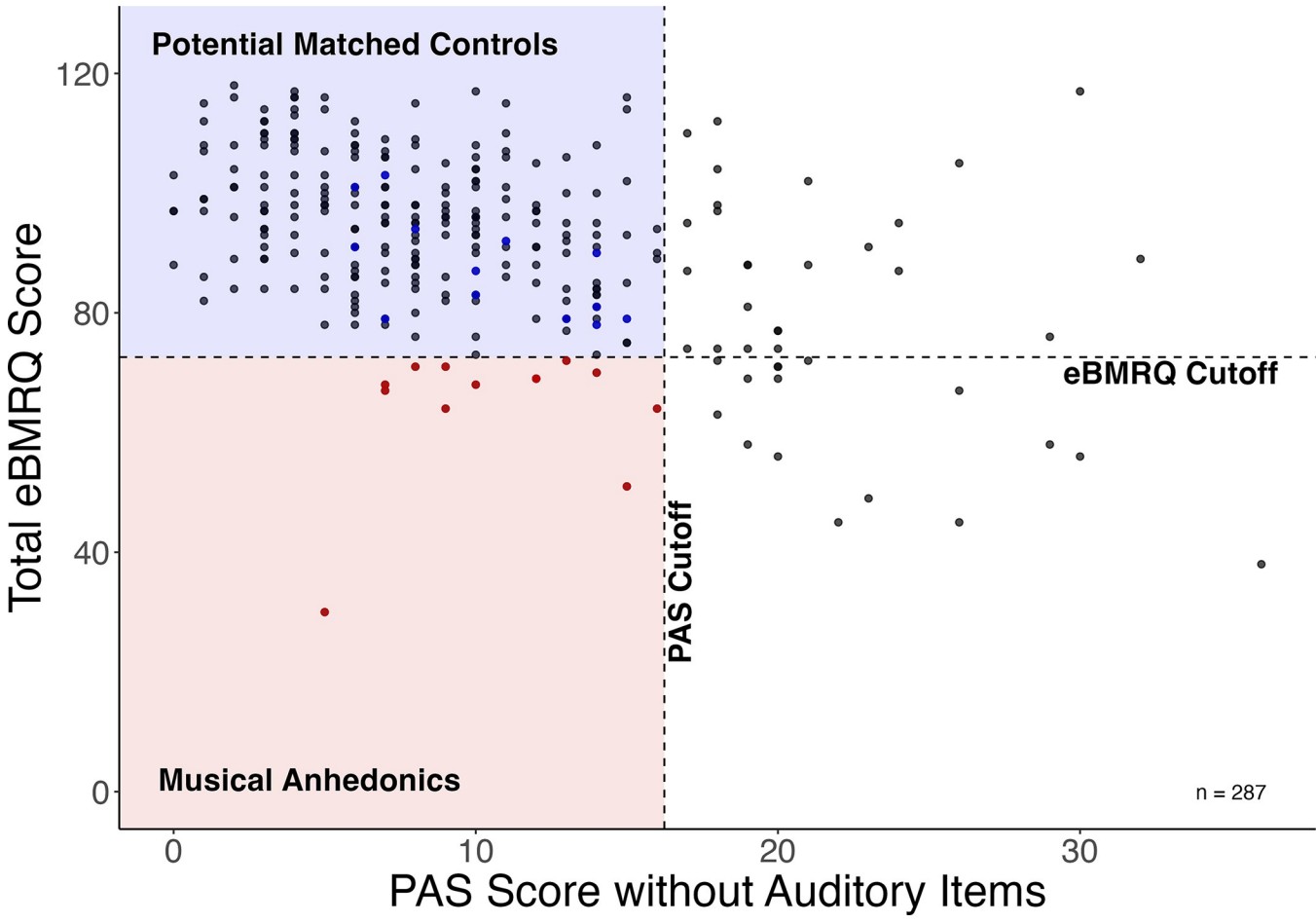

**Fig 1. Scatterplot showing the distribution of both eBMRQ and PAS scores without auditory items.** The vertical dotted line shows our threshold for general anhedonia whereas the horizontal dotted line shows our threshold for music reward sensitivity. Identified musical anhedonics are represented by the red dots and chosen matched controls are represented by the blue dots.

**Individual differences in music reward.** Using our entire sample (n = 287), we then tested our second hypothesis for differences in preferences for complexity as a function of individual differences in music reward sensitivity using continuous eBMRQ scores. To do so, we first constructed two linear mixed effects models with complexity tertile as a predictor variable for each rating type, with an interaction term for musical reward sensitivity (i.e. eBMRQ scores). We considered any significant interaction between eBMRQ and complexity tertile as evidence of differences in preference for complexity as a function of music reward sensitivity. Simple slopes relating eBMRQ to pleasure ratings within each complexity tertile, and pairwise contrast interactions comparing the strength of this relationship across tertiles, were tested using a Tukey test with the *emmeans* package.

As a final way to test for differences in the relationship between complexity and preference as a function of music reward sensitivity, we tested whether a linear or quadratic relationship best described the relationship between continuous perceived complexity values and ratings across our entire sample. For each of the two rating types, we constructed two linear mixed-effects models with complexity scores as a continuous predictor variable: one with just a linear term for perceived complexity, and another with an additional quadratic term for perceived complexity. We then compared the fit of these two models for each rating type using a

Likelihood Ratio test. To allow for model comparison, parameters in these models were estimated using maximum likelihood [44]. We then generated participant-level predictions using the best-fit model and isolated the complexity value of each participant's predicted maximal pleasure ratings (which represents the model-predicted optimal level of perceived complexity per participant). We then correlated this value with participants' eBMRQ score. This provided an additional way to investigate if individual differences in music reward sensitivity relate to preferences for levels of rhythmic complexity.

All reported models contained by-participant and by-item (i.e. stimuli) random intercepts, as well as by-participant random slopes for the effect of complexity tertile. Continuous variables were standardized before being entered into the models. Significance of fixed effects was determined using the Satterthwaite method to approximate the degrees of freedom with the *lmerTest* package [45] with an F-test.

## Results

### Preferences differ by rhythmic complexity and by music reward

Fig 2 shows mean ratings of pleasure and wanting to move for each tertile in perceived complexity for identified musical anhedonics and matched controls. For ratings of pleasure, there was a significant main effect of group ($F(1, 24) = 8.33$, $\eta_p^2 = 0.26$, $p = 0.001$), such that musical anhedonics reported less pleasure overall, and a significant main effect of complexity tertile such that the medium-complexity tertile was rated most pleasurable ($F(2, 18) = 5.42$, $\eta_p^2 = 0.38$, $p = 0.01$). However, there was no interaction between group and complexity tertile ($F(2, 37) = 0.13$, $\eta_p^2 = 0.001$, $p = 0.88$). There was a significant negative quadratic relationship, i.e. inverted-U curve, between pleasure ratings and complexity tertile across both groups (b = -0.77, $t(15) = -3.3$, $p = 0.048$). This inverted-U relationship was consistent within each group (musical anhedonics: b = -0.84, $t(28) = -2.8$, $p = 0.001$; matched controls: b = -0.7, $t(28) = -2.34$, $p = 0.03$) and did not differ between them (interaction contrast b = -0.14, $t(37) = -0.37$, $p = 0.71$).

For ratings of wanting to move, this revealed no significant effects of group ($F(1, 24) = 2.74$, $\eta_p^2 = 0.11$, $p = 0.10$), nor did it reveal a significant effect of complexity tertile ($F(2, 16) = 2.85$, $\eta_p^2 = 0.26$, $p = 0.67$). Furthermore, there was no group by complexity tertile interaction ($F(2, 34) = 0.41$, $\eta_p^2 = 0.02$, $p = 0.67$). We also found no significant quadratic relationship between move ratings and complexity tertile across both groups (b = -0.72, $t(13) = -1.93$, $p = 0.07$). This held for both groups (musical anhedonics: (b = -0.64, $t(19) = -1.56$, $p = 0.14$; matched controls: b = -0.80, $t(19) = -1.95$, $p = 0.07$) and did not differ between them (interaction contrast: b = 0.163, $t(58) = 0.47$, $p = 0.64$).

Results thus far show that musical anhedonics gave ratings below controls in pleasure but not in wanting to move, and the quadratic relationship with complexity was also observed for both groups in pleasure but not in move ratings. The two types of ratings could thus reflect different processes by which musical anhedonics evaluate complexity. In order to test whether the observed main effect of group membership on pleasure, but not wanting to move, ratings represents a meaningful effect of rating type on differences in music reward valuation, it would be necessary to test a three-way interaction between rating type (pleasure vs. wanting to move), group, and complexity tertile. However, our matched samples of n = 13 per group are likely underpowered to detect such interactions. Thus, we turned to using the full data from our larger sample to relate individual differences in reward sensitivity to preferences for complexity.

### Music reward sensitivity predicts preference for complexity

Considering the entire sample (n = 287), our model relating continuous eBMRQ as a predictor of pleasure ratings revealed a significant interaction between eBMRQ and complexity tertile (*F*

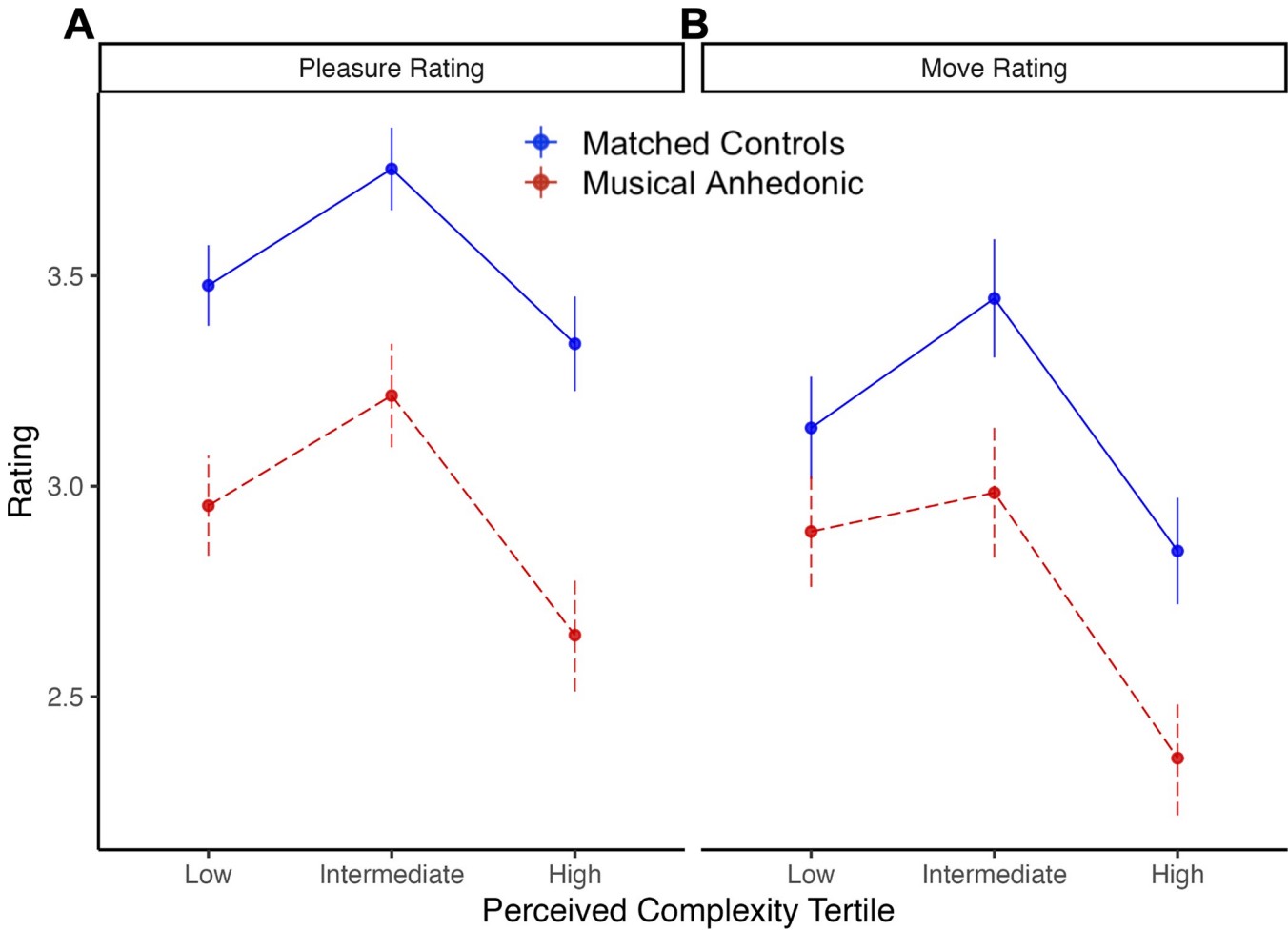

**Fig 2.** Mean ratings of pleasure **(A)** and wanting to move **(B)** for identified musical anhedonics and matched controls across the complexity tertiles. Error bars represent +/- 1 SE.

(2, 285) = 5.56, $\eta_p^2$ = 0.04, $p$ = 0.004). Specifically, there was no relationship between eBMRQ scores and pleasure ratings in response to stimuli in the low-complexity tertile (b = 0.004, $t$(285) = 1.47, $p$ = 0.14), but there was a significantly positive relationship between the two for the intermediate- (b = 0.01, $t$(285) = 2.96, $p$ = 0.003) and high-complexity tertiles (b = 0.01, $t$(285) = 4.92, $p$ < 0.001). Pairwise comparisons revealed that this relationship was significantly stronger in the high-complexity tertile compared to both the intermediate (b = 0.001, $t$(285) = 2.78, $p$ = 0.02) and low (b = 0.01, $t$(285) = 3.3, $p$ = 0.003). There was no significant difference in the relationship between eBMRQ scores and pleasure ratings between intermediate- and low-complexity stimuli (b = 0.003, $t$(285) = 1.62, $p$ = 0.24; see Fig 3A).

For move ratings, this analysis revealed a significant interaction between eBMRQ and complexity tertile ($F$(2, 382) = 3.04, $\eta_p^2$ = 0.04, $p$ = 0.049). There was no significant relationship between eBMRQ scores and move ratings in the low-complexity tertile (b = 0.004, $t$(286) = 1.57, $p$ = 0.12), a significant relationship between the two for the intermediate-complexity tertile (b = 0.007, $t$(297) = 3.06, $p$ = 0.002), and a significant relationship in the high-complexity tertile (b = 0.013, $t$(285) = 4.24, $p$ < 0.001). Furthermore, pairwise comparisons of these relationships revealed that the relationship was significantly stronger in the high-complexity tertile compared to the low-complexity tertile (b = -0.007, $t$(285) = -2.46, $p$ = 0.04). There was no

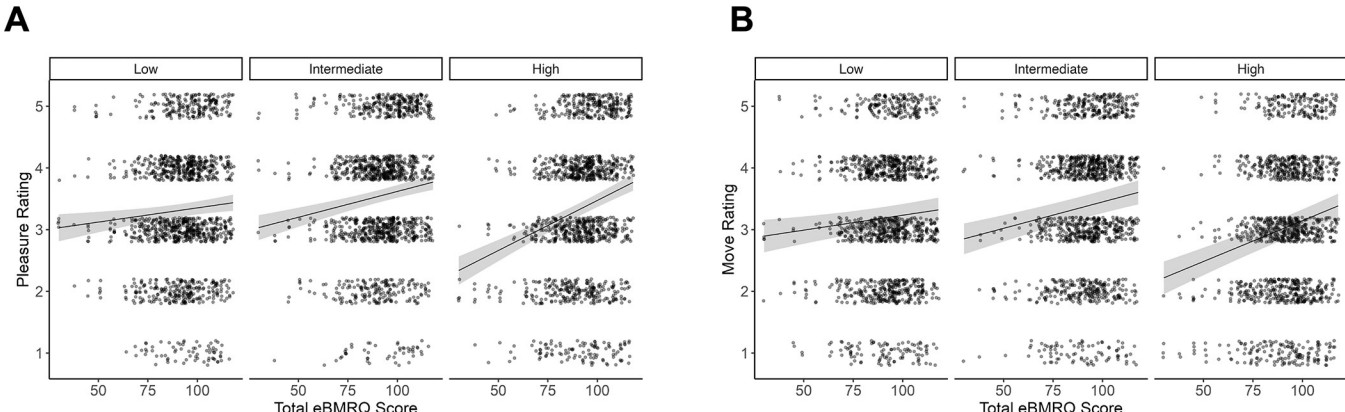

**Fig 3.** Model-predicted pleasure (**A**) and wanting to move (**B**) ratings as a function of participants' total eBMRQ scores for each of the three complexity tertiles. Shaded regions represent +/- 1 SE of the model predictions. Points represent raw participant ratings, which are jittered slightly for visualization.

significant difference in the relationships between eBMRQ scores and move ratings between high- and intermediate-complexity tertiles (b = -0.004, $t(325)$ = -1.67, $p$ = 0.22), or between low- and intermediate-complexity tertiles (b = 0.003, $t(618)$ = 1.56, $p$ = 0.27; see Fig 3B).

To further clarify the three-way relationship between ratings type, complexity, and musical reward sensitivity, we ran an additional follow-up model with ratings as a function of three factors: rating type (pleasure and wanting to move), eBMRQ scores, and complexity tertile, to investigate if the relationships reported above differed across pleasure and move ratings. Results of this model are shown in Table 2. This model revealed a main effect of rating type, such that ratings of pleasure were higher than ratings of wanting to move, and a main effect of eBMRQ, such that participants with higher eBMRQ score made higher ratings. There was also a significant interaction between complexity tertile and eBMRQ, such that people with higher eBMRQ scores gave higher ratings for high-complexity items: the relationship between eBMRQ and ratings was stronger in response to the high-tertile complexity compared to both the intermediate- (b = 0.005, t(285) = 2.38, p = 0.047) and low- (b = 0.009, t(285) = 3.06, p = 0.007) tertile complexity stimuli (Fig 3). However, there was no two-way interaction between eBMRQ scores and rating type, nor was there a three-way interaction between eBMRQ scores, complexity tertile, and rating type. Thus, including the variable of complexity reveals the effect of individual differences in musical reward on both types of ratings–such that high-complexity stimuli were especially pleasurable and especially movement-inducing to highly reward-sensitive individuals.

**Table 2.** *F*- and *p*-values of fixed effects from a linear mixed effects model testing a three-way interaction between rating type (move or pleasure), complexity tertile, and eBMRQ scores on both pleasure and move ratings.

| Term | F (numerator df, denominator df) | p |
|---|---|---|
| Rating Type | 114.92 (1, 7731) | < 0.001 |
| Complexity Tertile | 1.16 (2, 13) | 0.34 |
| eBMRQ Score | 14.75 (1, 285) | < 0.001 |
| Rating Type X Complexity Tertile | 8.57 (2, 7731) | < 0.001 |
| Rating Type X eBMRQ Score | 0.48 (1, 7731) | 0.49 |
| Complexity Tertile X eBMRQ Score | 4.69 (2, 285) | 0.001 |
| Rating Type X Complexity Tertile X eBMRQ Score | 0.73 (2, 7731) | 0.48 |

**Table 3. Parameter estimates and model fit indices for models treating either pleasure (A) or wanting to move ratings (B) as a function of continuous perceived complexity (for both linear and quadratic models).**

| A) Pleasure Ratings | | | | |
|---|---|---|---|---|
| **Model:** | **Linear** | | **Quadratic** | |
| **Effect** | b | p | b | p |
| Perceived Complexity | 0.005 | 0.93 | 0.33 | 0.93 |
| Perceived Complexity$^2$ | | | -5.34 | 0.13 |
| **R$^2$ (Conditional, Marginal):** | 0.00003, 0.436 | | 0.007, 0.441 | |
| **B) Wanting to Move Ratings** | | | | |
| **Model:** | **Linear** | | **Quadratic** | |
| **Effect** | b | p | b | p |
| Perceived Complexity | -0.07 | 0.47 | -4.33 | 0.47 |
| Perceived Complexity$^2$ | | | -6.51 | 0.28 |
| **R$^2$ (Conditional, Marginal):** | 0.004, 0.432 | | 0.014, 0.441 | |

Finally, since the complexity measures from the present stimuli were chosen to reflect points along a continuum of perceived complexity rather than three distinct tertiles, we further evaluated quadratic versus linear models for both rating types. Likelihood ratio tests revealed that a quadratic model fit better than a linear model for both pleasure ($\chi^2(4) = 17.92$, $p = 0.001$) and wanting to move ratings ($\chi^2(4) = 14.79$, $p = 0.005$) (see Table 3 for model fits and parameter estimates and Fig 4A and 4D for plotted model predictions). Further, complexity values of participants' peak predicted ratings from these quadratic models were significantly positively correlated to eBMRQ scores (pleasure ratings: $r(285) = 0.2$, $p < 0.001$; wanting to move ratings: $r(285) = 0.15$, $p = 0.01$). In other words, participants who are highly sensitive to music reward prefer higher complexity compared to those with lower sensitivity to music reward, which results in a shift in the apex of their individual inverted-U curve relating musical preference to complexity. This shift in peak of the inverted-U curve across participants with different music-reward sensitivity is observable in Fig 4C and 4F, which show participant-level predictions from these quadratic models for relatively low, intermediate, and high scorers on the eBMRQ in our sample.

### Sensorimotor subscale is strongest predictor of move ratings

We also explored the relationships of individual subscale scores of the eBMRQ with both pleasure and move ratings by constructing linear mixed-effects models relating ratings to all eBMRQ subscales. These models contained by-participant and item random intercepts, as well as random by-participant effects for each subscale. For pleasure ratings, no specific subscale was a significant predictor above the influence of others. For move ratings, only the sensorimotor subscale emerged as a significant predictor (see Table 4 for results of both models).

To formally investigate if this relationship with the sensorimotor subscale differed between the two ratings, we modeled the sensorimotor subscale as a function of both ratings with an interaction term for rating type. This model again revealed a main effect of rating type ($F(1, 8307) = 99.08$, $\eta_p^2 = 0.21$, $p < 0.001$), such that pleasure ratings were higher than move ratings. Further, while the sensorimotor subscale scores were significantly associated with ratings ($F(1, 130) = 5.67$, $\eta_p^2 = 0.05$, $p = 0.02$), this association did not differ between rating types (interaction effect: $F(1, 8307) = 0.47$, $\eta_p^2 = 0.002$, $p = 0.49$). This suggests that while this subscale is uniquely associated with move ratings among other subscales of the eBMRQ, there is no significant difference in its relationship between move and pleasure ratings.

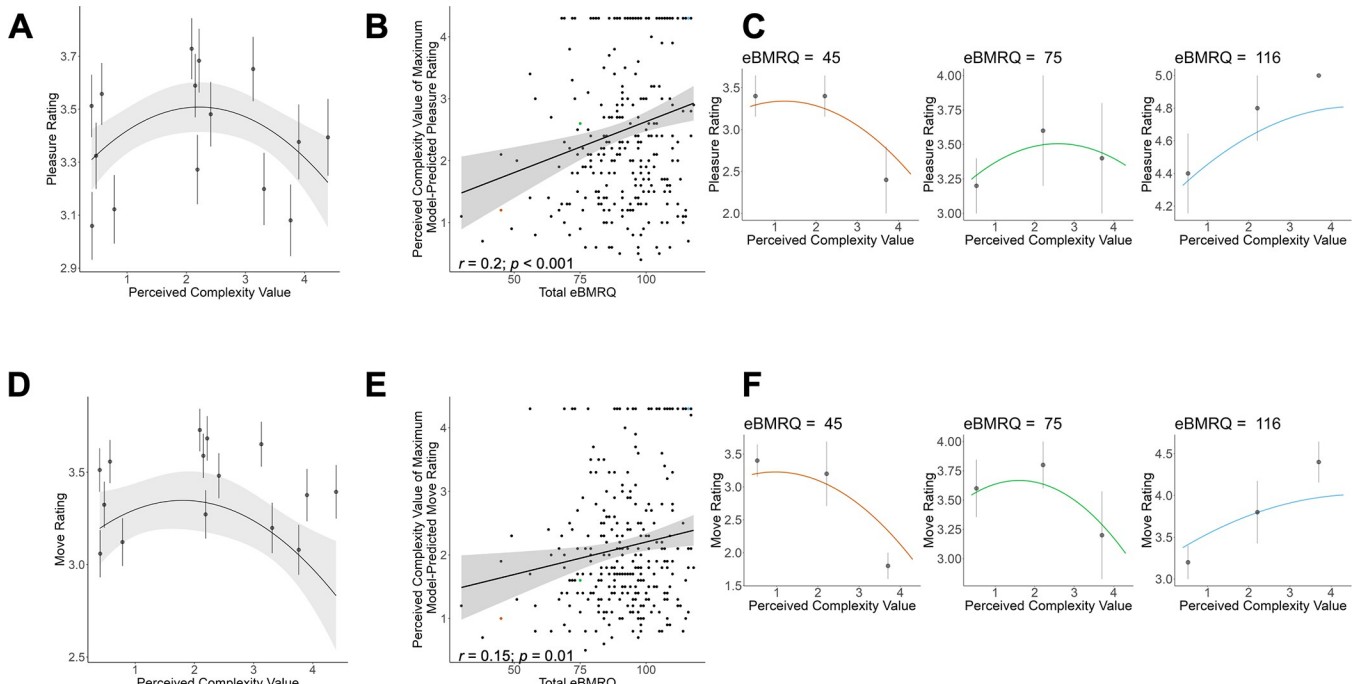

**Fig 4.** Model-predicted pleasure **(A)** and wanting to move **(D)** ratings as a function of perceived complexity values, treated continuously, from best-fit quadratic models. Points represent mean ratings for each stimulus, with associated error bars representing +/- 1 SE and shaded regions represent +/- 1 SE of the model predictions. Participant-level predictions were generated from these models, and the perceived value of each participant's predicted maximum rating of pleasure **(B)** and wanting to move **(E)**, representing their individual optimal level of complexity for both ratings, are plotted as a function of their overall eBMRQ scores. Panels **C** and **F** show examples of these participant-level predictions for individuals who scored relatively low (shown in red in panels **B** and **E**; leftmost plots), average (shown in green in panels **B** and **E**; center plots), and high (shown in blue in panels **B** and **E**; rightmost plots) on the eBMRQ in our sample. Points in the plots of panels **C** and **F** represent the mean for each participant in each tertile complexity group (i.e. low, intermediate, and high complexity) and associated error bars represent +/- 1 SE. *Note*: there is no error bar in the rightmost plot of panel **C** because this participant rated all high complexity stimuli a "5".

### Musical Ear Test predicts preference for complexity

To directly investigate the effects of rhythm perception on the experience of groove, we constructed a linear mixed-effects model relating both pleasure and move ratings as a function of MET rhythmic score, with complexity tertile as an interaction term. This model had the same random effects structure as previous models. The model revealed a significant interaction between MET rhythmic score and complexity tertile ($F(2, 336) = 7.65$, $\eta_p^2 = 0.04$, $p < 0.001$) for pleasure ratings: MET rhythmic scores were significantly positively related to pleasure ratings in both the intermediate (b = 0.01, $t(291) = 2.12$, $p = 0.03$) and high (b = 0.02, $t(285) = 2.78$, $p = 0.001$) complexity tertiles, but not in the low-complexity tertile (b = -0.01, $t(2.86) = -0.71$, $p = 0.48$). Pairwise comparisons of these relationships revealed that this relationship was significantly stronger in the high-complexity tertile compared to the low-complexity tertile (b = 0.03, $t(285) = 3.32$, $p = 0.003$) and in the intermediate-complexity tertile compared to the low-complexity tertile (b = 0.02, $t(440) = 3.62$, $p = 0.001$). There was no significant difference in this relationship within the intermediate-complexity tertile compared to within the high-complexity tertile (b = 0.01, $t(301) = 1.19$, $p = 0.46$).

For move ratings, this again revealed a significant MET rhythmic score by complexity tertile interaction ($F(2, 381) = 6.13$, $\eta_p^2 = 0.03$, $p = 0.002$). Simple slopes within each tertile complexity level of MET rhythmic scores did not reach statistical significance (low: b = -0.01, $t(287) = -1.37$, $p = 0.17$; intermediate: b = 0.01, $t(296) = 1.22$, $p = 0.22$; high: b = 0.01, $t(285) = 1.8$,

**Table 4. Estimated standardized beta coefficients, along with associated *t*- and *p*-values, from models treating pleasure (A) and wanting to move (B) ratings as a function of the six subscales of the eBMRQ. Because we were concerned about multicollinearity in these models due to the likelihood that these subscale scores were highly correlated, we also included variance inflation factors (VIF) for each predictor variable, which indicate low multicollinearity among variables across both models (i.e., none meet the typical >10 threshold, which suggests high multicollinearity among predictors [46]).**

| A) Pleasure Ratings | | | | |
|---|---|---|---|---|
| **eBMRQ Subscale** | *b* | *t (df)* | *p* | Variance Inflation Factor (VIF) |
| Sensorimotor | 0.02 | 0.48 (79) | 0.64 | 1.88 |
| Music Seeking | 0.07 | 1.29 (114) | 0.20 | 2.75 |
| Emotional Regulation | -0.02 | -0.39 (138) | 0.69 | 1.96 |
| Mood Regulation | 0.05 | 0.98 (123) | 0.33 | 2.62 |
| Social Reward | -0.01 | -0.31 (118) | 0.76 | 2.23 |
| Absorption | 0.08 | 1.85 (125) | 0.07 | 1.41 |
| **B) Wanting to Move Ratings** | | | | |
| **eBMRQ Subscale** | b | *t (df)* | *p* | Variance Inflation Factor (VIF) |
| Sensorimotor | 0.08 | 2.35 (89) | 0.02 | 1.33 |
| Music Seeking | 0.09 | 1.87 (95) | 0.06 | 2.58 |
| Emotional Regulation | -0.05 | -1.26 (114) | 0.21 | 1.7 |
| Mood Regulation | 0.06 | 1.14 (100) | 0.26 | 2.25 |
| Social Reward | -0.05 | -1.19 (147) | 0.24 | 1.87 |
| Absorption | 0.04 | 0.93 (103) | 0.36 | 1.33 |

*p* = 0.07). However, pairwise comparisons indicate that the relationship between MET scores and move ratings was stronger in the intermediate-complexity tertile compared to the low-complexity tertile (b = 0.02, *t*(670) = 3.24, *p* = 0.004) and in the high-complexity tertile compared to the low-complexity tertile (b = 0.02, *t*(285) = 2.94, *p* = 0.001), suggesting that MET rhythmic scores best predict move ratings in stimuli with higher perceived complexity.

To formally test whether MET scores better predicted pleasure ratings or wanting to move ratings, we ran an additional three-way (rating type X MET rhythmic score X complexity tertile) interaction model with ratings (both pleasure and move) as the outcome variable. Results from this model are shown in Table 5. There was a main effect of rating type: pleasure ratings were higher than move ratings, as reported above. There was a significant two-way interaction between complexity tertile and MET rhythmic score, consistent with results reported above. There was no significant three-way interaction with complexity tertile, suggesting that the relationship between complexity and MET rhythmic score was indifferentiable by rating type. Critically, this model revealed a significant two-way interaction between rating type and MET

**Table 5. *F*- and *p*-values of fixed effects from a linear mixed effects model testing a three-way interaction between rating type (move or pleasure), complexity tertile, and MET Rhythmic scores on both pleasure and move ratings.**

| Model Term | *F* *(numerator df, denominator df)* | *p* |
|---|---|---|
| Rating Type | 115 (1, 7731) | < 0.001 |
| Complexity Tertile | 1.16 (2, 13) | 0.34 |
| MET Rhythmic Score | 1.5 (1, 285) | 0.22 |
| Rating Type X Complexity Tertile | 8.57 (2, 7731) | < 0.001 |
| Rating Type X MET Rhythmic Score | 4.91 (2, 285) | 0.03 |
| Complexity Tertile X MET Rhythmic Score | 7.88 (2, 285) | < 0.001 |
| Rating Type X Complexity Tertile X MET Rhythmic Score | 0.15 (2, 7731) | 0.86 |

rhythmic score, such that MET rhythmic scores were significantly more related to pleasure than to move ratings (contrast: b = 0.001, *t*(7731) = -2.22, *p* = 0.03).

## Discussion

The relationship between aesthetic pleasure and movement is of intense interest in perception and performance, especially within empirical aesthetics and music cognition. Here we examine how reward sensitivity affects the relationship between stimulus complexity and both pleasure and movement in the domain of music. Using a newly-released set of naturalistic rhythmic stimuli, we replicate previous reports of the inverted-U-shaped relationship between complexity and subjective ratings, more for pleasure than for wanting to move as discussed below. Musical anhedonics' ratings were lower than matched controls on pleasure but not on move ratings. Considering the entire sample, we found that the inflection point of the inverted-U for both pleasure and wanting to move was predicted by individual differences in rhythm discrimination and reward sensitivity: more reward-sensitive individuals and those with higher rhythm discrimination skills tended to prefer higher perceived complexity. Results advance our understanding of the relationship between pleasure and movement, by relating individual differences in reward sensitivity and rhythm discrimination to preference for complexity.

We found that musical anhedonics do experience groove, as defined by higher pleasure for syncopated (medium-complexity) rhythmic stimuli. At the same time, we observed dissociations between pleasure and wanting to move in musical anhedonia: while musical anhedonics' pleasure ratings were lower than matched controls overall, their wanting to move ratings were not significantly lower. Part of this may be explained by the fact that the process of selecting musical anhedonics and matched controls resulted in rather small sample sizes that were underpowered to detect a between-group difference. Turning to an individual-differences approach, rather than a case-control approach, allowed us to expand our usable sample size to the entire sample (n = 287 overall rather than 13 per group), thus revealing significant relationships between preference for complexity and rhythm discrimination skills as well as reward sensitivity.

Directly comparing our two outcome variables of pleasure ratings and move ratings, we saw that the pleasure ratings here are higher than the move ratings. This differs from prior studies that have shown that pleasure and move ratings are statistically indistinguishable and show similar relationships with complexity and syncopation [5]. Part of this difference, i.e. higher pleasure ratings than move ratings, may reflect our choice of stimuli: the present stimulus set is newly published and validated by perceived complexity ([23] as derived from [47]); thus fewer research groups have used them. The present stimuli are all audio reconstructions of expert transcriptions of 8-bar extracts of original recordings by expert drummers, whereas stimuli from previous studies are not derived from originally recorded popular music, but were composed by the researchers to show extreme (high or low) levels of syncopation. Furthermore the present stimuli added reverberation in their reconstructions of the expert drum patterns, which increased acoustic liveness of the percept, and thus the perceived ecological validity of the experiment for the listener. While researcher-composed stimuli are effective in establishing the landmark findings that relate the scientifically generalizable concept of syncopation and rhythmic complexity to the pleasurable urge to move, stimuli that are more ecologically valid and that maximize the enjoyment of the listener may be effective at bridging the gap between lab-based research and real-world applications [48], such as the design and implementation of music-based interventions that tap into pleasure and movement mechanisms for movement disorders such as Parkinson's Disease [3, 49].

Our results suggest that pleasure and wanting to move are driven by at least partly separable mechanisms. While groove is defined as a pleasurable urge to move [2], the urge to move only

captures part of what can make musical sounds pleasurable. Breaking down the individual-difference measures into their subscales was informative in this regard, as the subscales are designed to capture different facets of musical reward: our results show that the sensorimotor subscale of the eBMRQ is most predictive of move ratings, whereas the rhythm subscale of the MET is most predictive of pleasure ratings. The relationship between sensorimotor eBMRQ and move ratings is expected: those who identify that music makes them want to move are more likely to give higher wanting-to-move ratings. How pleasure ratings relate to MET rhythm subscale is less clear: As these results are correlational, they are unable to establish the direction of causality between rhythmic discrimination and pleasure. Nevertheless, as the MET is designed to assess discrimination skills, the present results may suggest that participants who have better rhythm discrimination skills may be better able to appreciate the pleasures of highly rhythmically complex stimuli. Alternately, individuals who experience complex drum beats as more pleasing may be more motivated to perform on the rhythm discrimination task.

Another important part of our findings is that preference for complexity scaled with eBMRQ. Those with higher reward sensitivity preferred stimuli with higher perceived complexity, such that very highly reward-sensitive individuals (cf. "musical hyperhedonics", [16]) preferred most complex stimuli, resulting in a monotonic increase of pleasure with complexity, rather than an inverted-U curve. Individual differences in the preference for complexity have been examined in prior work: For stimuli that were composed to vary in harmonic complexity, musically trained individuals showed a rightward skew of the inverted-U relative to their counterparts who reported no musical training [7]. Using excerpts of digitized music from real, precomposed sources, it was also previously shown that musicality as assessed by Gold-MSI predicted the skew and kurtosis of the inverted-U curve relating preference to musical complexity as quantified by entropy and information content [50]. Here, we provide a conceptual replication of these previous results, using naturalistic drum pattern stimuli and relating perceived complexity to preference for individuals who vary along the continuum of musical reward sensitivity.

Perceived complexity offers a theoretically motivated and empirically validated continuum of complexity measures, while being relatively naturalistic and ecologically valid [23]. Due to time constraints for participants in the present study, we chose to collect ratings from a subset of 15 stimuli curated from the database of 40 drumbeat stimuli, representing the high, medium, and low categories of a fuller continuum of perceived complexity. Effectively, this undersampling of the inverted-U curve offers a replication of previous work using composed stimuli [5, 9]. Thus, while we were unable to capture the full continuum of perceived complexity, this approach allows us to more rigorously examine previous demonstrations of the relationship between complexity and aesthetics using a novel set of stimuli. Future work may seek to expand our sampling strategies to quantify the shape of the relationship between complexity, pleasure, and the urge to move, and extend them into more naturalistic contexts.

Taken together, the current study finds the experience of groove to be related to perceived rhythmic complexity. Preference for rhythmic complexity is related to the sensorimotor aspect of individual differences in musical reward, and to rhythm discrimination ability. Results inform our understanding of the relationship between aesthetic pleasure and movement, and may be useful for future designs of active music-based interventions, such as those that involve walking or dancing to music for a variety of neurological disorders.

## Supporting information

**S1 File.**
(DOCX)

## Acknowledgments

Supported by NSF-CAREER 1945436, NSF-BCS 2240330, NIH R01AG078376 and R21AG075232 to PL, NSF GRFP DGE-1938052 to NK, and internal seed funding from Northeastern University.

## Author Contributions

**Conceptualization:** Peter Benson, Nicholas Kathios, Psyche Loui.

**Data curation:** Peter Benson, Nicholas Kathios, Psyche Loui.

**Formal analysis:** Peter Benson, Nicholas Kathios, Psyche Loui.

**Funding acquisition:** Nicholas Kathios, Psyche Loui.

**Investigation:** Peter Benson, Nicholas Kathios, Psyche Loui.

**Methodology:** Peter Benson, Nicholas Kathios, Psyche Loui.

**Project administration:** Peter Benson.

**Resources:** Nicholas Kathios, Psyche Loui.

**Software:** Peter Benson, Nicholas Kathios.

**Supervision:** Nicholas Kathios, Psyche Loui.

**Validation:** Psyche Loui.

**Visualization:** Peter Benson, Nicholas Kathios, Psyche Loui.

**Writing – original draft:** Peter Benson, Nicholas Kathios, Psyche Loui.

**Writing – review & editing:** Peter Benson, Nicholas Kathios, Psyche Loui.

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
