## [Decision Letter · Decision Letter 0]

17 Mar 2024

Predictive Coding in Musical Anhedonia: A Study of Groove

PONE-D-24-02487

Dear Dr. Loui,

We’re pleased to inform you that your manuscript has been judged scientifically suitable for publication and will be formally accepted for publication once it meets all outstanding technical requirements.

Kind regards,

Maja Vukadinovic

Academic Editor

PLOS ONE

Reviewers' comments:

Reviewer's Responses to Questions

**Comments to the Author**

1. Is the manuscript technically sound, and do the data support the conclusions?

Reviewer #1: Yes

Reviewer #2: Yes

2. Has the statistical analysis been performed appropriately and rigorously? 

Reviewer #1: Yes

Reviewer #2: Yes

3. Have the authors made all data underlying the findings in their manuscript fully available?

Reviewer #1: Yes

Reviewer #2: Yes

4. Is the manuscript presented in an intelligible fashion and written in standard English?

Reviewer #1: Yes

Reviewer #2: Yes

5. Review Comments to the Author

Reviewer #1: This paper sets out an investigation regarding musical anhedonics' experience of groove in drum patterns. Building on earlier work, by Witek and others, and using a relatively new ecologically-strong stimulus set, this study interrogates responses on pleasure and desire to move by a sample of 287 online participants, a subset of which (n=13) were adjudged musically anhedonic. The results found the expected inverted-U pattern for responses relative to stimulus complexity, for anhedonics and others. Anhedonics do report both pleasure and desire to move but at lower levels than others (as expected). The most interesting findings are probably a) pleasure and desire to move are seem driven by at least partially separable mechanisms and b)from the individual-differences subset of the analyses, degree of reward sensitivity correlates with preference for more complex stimuli.

Moving from summary to suggestions: This is a clear and well-written paper and therefore few of the suggestions I would usually make are indicated. I feel that the paper comes off as statistical and technical rather than intellectually-driven. The paper places a burden on the reader to constantly track the train of argumentation, as the "decision tree" of question, statistical test, result, further questions <...> proceeds. This is foretold in the Abstract, where findings start halfway through, and the distinctive contributions appear only in the last two sentences. I am not primarily a statistician (but am glad I know as much R as I do, while reading this paper) and felt that the bulk of the paper emphasized what tests were being performed rather than what questions were being pursued and why. Generally a new sub-inquiry was introduced with only one or two motivating sentences, followed by several dense sentences of test types and outputs, and then a sentence or so of interpretation. It would, in my estimation, help readers if the chain of reasoning were set out more fully and distinctly throughout the paper (if this was a conference talk this might be accomplished by an outline that is gradually filled in --"ungreyed," perhaps--question by question and test by test). This would also relieve some of the pressure on the Discussion to make things clear.

Beyond this there are only two things I wish to mention. The first is probably an arrtifact of the online submission process: Figure 4 lacks headings for panels A, B, and C. The second is that I question the causal directionality of the Discussion's statement "the present results suggest that individuals who rate complex drum beats as more pleasing may be more motivated to perform on the rhythm discrimination task," which at the least could use more explication.

Reviewer #2: The article highlights its significant contributions, from its theoretical framework, research procedures, statistical analysis of results and discussions, confirming its potential impact in the field of music research. It is possible to appreciate the depth of the research and the clarity of presentation of all topics and issues. This work deserves to be published because it represents a significant contribution to the academic framework and contains valuable knowledge for researchers and in this specific line of action of Music Psychology.

6. PLOS authors have the option to publish the peer review history of their article (what does this mean?). If published, this will include your full peer review and any attached files.

Reviewer #1: No

Reviewer #2: **Yes: **Ruth Nayibe Cárdenas Soler

---

## [Editor Report · Acceptance letter]

8 Apr 2024

PONE-D-24-02487 

PLOS ONE

Dear Dr. Loui, 

I'm pleased to inform you that your manuscript has been deemed suitable for publication in PLOS ONE. Congratulations! Your manuscript is now being handed over to our production team.

Kind regards, 

on behalf of

Dr. Maja Vukadinovic 

Academic Editor

PLOS ONE